# Establishing Trustworthiness: Rethinking Tasks and Model Evaluation

**Robert Litschko**[1,3*]     **Max Müller-Eberstein**[2*]   **Rob van der Goot**[2]
**Leon Weber**[1,3]     **Barbara Plank**[1,2,3]

[1] MaiNLP, Center for Information and Language Processing, LMU Munich, Germany
[2] Department of Computer Science, IT University of Copenhagen, Denmark
[3] Munich Center for Machine Learning (MCML), Munich, Germany
{rlitschk, leonweber, bplank}@cis.lmu.de   {mamy, robv}@itu.dk

## Abstract

Language understanding is a multi-faceted cognitive capability, which the Natural Language Processing (NLP) community has striven to model computationally for decades. Traditionally, facets of linguistic intelligence have been compartmentalized into tasks with specialized model architectures and corresponding evaluation protocols. With the advent of large language models (LLMs) the community has witnessed a dramatic shift towards general purpose, task-agnostic approaches powered by generative models. As a consequence, the traditional compartmentalized notion of language tasks is breaking down, followed by an increasing challenge for evaluation and analysis. At the same time, LLMs are being deployed in more real-world scenarios, including previously unforeseen zero-shot setups, increasing the need for trustworthy and reliable systems. Therefore, we argue that it is time to rethink what constitutes tasks and model evaluation in NLP, and pursue a more holistic view on language, placing trustworthiness at the center. Towards this goal, we review existing compartmentalized approaches for understanding the origins of a model's functional capacity, and provide recommendations for more multi-faceted evaluation protocols.

> "Trust arises from knowledge of origin as well as from knowledge of functional capacity."

*Trustworthiness - Working Definition*
*David G. Hays, 1979*

## 1 Introduction

Understanding natural language requires a multitude of cognitive capabilities which act holistically to form meaning. Modeling this ability computationally is extremely difficult, thereby necessitating a compartmentalization of the problem into *isolated tasks* which are solvable with available methods and resources (Schlangen, 2021). Undoubtedly

---
* Equal contribution.

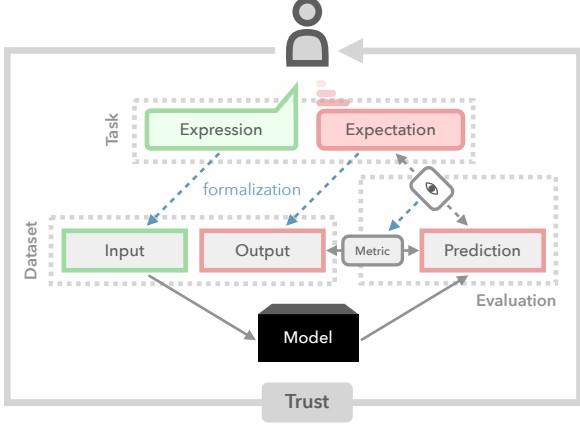

Figure 1: **Contemporary NLP Paradigm** with language tasks formalized as datasets for which models produce predictions. Recent LLMs break down this compartmentalization (dashed lines), impacting all stages of the cycle. We argue that establishing *trust* requires rethinking every facet of this framework, as formalization and evaluation become increasingly difficult.

as of late 2022, we are witnessing a paradigm shift: Powerful LLMs, in the form of instruction-tuned, prompt-based generative models such as ChatGPT and GPT-4 (Wei et al., 2022a; Touvron et al., 2023b; Taori et al., 2023; OpenAI, 2023; Bubeck et al., 2023, *inter alia*), have found widespread adoption reaching far beyond the NLP community. Part of this success story is the casting of heterogeneous NLP tasks into sequence-to-sequence tasks (Raffel et al., 2020; Sanh et al., 2022; Wang et al., 2022b); which in turn enables extreme multi-task learning, and cross-task transfer learning.

This is in stark contrast to the traditional compartmentalized NLP paradigm (visualized in Figure 1), wherein a human-motivated language task with an input *expression* and an output *expectation* is clearly formalized into a *dataset* with machine-readable inputs and outputs. Both feature design and model development are highly task-specific—often manually curated. Paired with evaluation protocols for comparing model predictions with

human expectations via formalized metrics or qualitative judgement, this general methodology has been widely adopted and trusted.[1] However, with contemporary LLMs this compartmentalization is breaking down—having severe impacts on all stages of the cycle. Therefore, a persistent and critical question regains importance: *How can trust be established between the human and the model?*

As early as 44 years ago, Hays (1979) offers an attempt and provides a definition of *trustworthiness* (cf. quote). Today, the topic of trustworthiness is an ongoing discussion deserving special attention (Baum et al., 2017; Eisenstein, 2022; Clarke et al., 2023). We argue that to establish trust, it is time to rethink how we deal with tasks and their evaluation. Why now? It is getting increasingly hard to predict a priori when we can expect models trained on web-scale data to work well. Were we to live in a hypothetical world with full knowledge of origin and functional capacity, then each task instance could be routed to the right model(s) to not only tap into the LLMs' full potential, but to also enable trust in their predictions. Today, the absence of this knowledge is directly linked to our lack of trust in deploying models in real-world scenarios.

In this position paper, we synthesize contemporary work distributed throughout different subfields of NLP and ML into a conceptual framework for trust, guided by Hays (1979)'s definition and centered around *knowledge facets* as a guiding principle for all aspects of the model development and evaluation cycle. We outline high-level desiderata (§2), and suggest directions on how to gain trust, by providing starting points of facets (§3) aimed to stipulate uptake and discussion. In §4 we discuss how trustworthiness relates to user trust.

## 2 Desiderata for Trustworthy LLMs

LLMs today pose a conundrum: They are seemingly universally applicable, having high functional capacity, however, the larger the model, the less we appear to know about the origins of its capabilities. How did we get here, which aspects contribute to trustworthiness, and what did we lose on the way? In the following, we aim to provide a brief history of central trust desiderata (**D1-4**), discussing how our knowledge of functional capacity and its origins has changed over time.

---

[1]While not without deficiencies, evaluation protocols were arguably more heterogeneous and established than today w.r.t. quantitative/qualitative evaluation, human judgements etc.

**D1. Knowledge about Model Input.** In the beginnings of NLP, researchers followed strict, task-specific formalizations and had precise control over which "ingredients"[2] go into model training and inference (i.e., manual feature engineering). Neural models have caused a shift towards *learning* representations, improving performance at the cost of interpretability. While analogy tasks (Mikolov et al., 2013) have enabled analyses of how each word-level representation is grounded, contemporary representations have moved to the subword level, and are shared across words and different languages, obscuring our knowledge of the origin of their contents, and requiring more complex lexical semantic probing (Vulić et al., 2020, 2023). This is amplified in today's instruction-based paradigm in which tasks are no longer formalized by NLP researchers and expert annotators but are formulated as natural language expressions by practitioners and end users (Ouyang et al., 2022). The cognitive process of formalizing raw model inputs into ML features has been incrementally outsourced from the human to the representation learning algorithm, during which we lose knowledge over functional capacity.

**D2. Knowledge about Model Behaviour.** In the old compartmentalized view of NLP, higher-level tasks are typically broken down into pipelines of subtasks (Manning et al., 2014), where inspecting intermediate outputs improves our knowledge about model behaviour. Recently however, LLMs are usually trained on complex tasks in an end-to-end fashion (Glasmachers, 2017), which makes it more difficult to expose intermediate outputs and analyze error propagation. Over time we have gained powerful black-box models, but have lost the ability to interpret intermediate states and decision boundaries, thus increasing uncertainty and complexity. Because as of today, we cannot build models that always provide factually correct, up-to-date information, we cannot trust to employ these models at a large scale, in real-world scenarios, where reliability and transparency are key. In this regard, pressing questions are e.g., how *hallucination* and *memorization* behaviour can be explained (Dziri et al., 2022; Mallen et al., 2023), how models behave when trained on many languages (Conneau et al., 2020; Choenni et al., 2023), what internal features are overwritten when trained on differ-

---

[2]We refer to ingredients as explicit inputs and LLM's parametric knowledge (De Cao et al., 2021; Mallen et al., 2023).

ent tasks sequentially (*catastrophic forgetting*; e.g., McCloskey and Cohen, 1989; French, 1999), how to improve models' ability to know when they do not know (*model uncertainty*; e.g., Li et al., 2022a), or how do LLMs utilize skills and knowledge distributed in their model parameters.

**D3. Knowledge of Evaluation Protocols.** The emergence of LLMs has raised the question of how to evaluate general-purpose models. Many recent efforts have followed the traditional NLP evaluation paradigm and summarized LLM performance into evaluation metrics across existing benchmark datasets (Sanh et al., 2022; Wang et al., 2022b; Scao et al., 2022; Wei et al., 2022a; Touvron et al., 2023a). This estimates LLM performance for tasks covered by the benchmark dataset and thus establishes trust when applying the model to the same task. However, the situation is different when LLMs are used to solve tasks outside of the benchmark, which is often the case for real-world usage of LLMs (Ouyang et al., 2022). Then, the expected performance becomes unclear and benchmark results become insufficient to establish trust. One proposal to solve this issue is to evaluate on a wide variety of task-agnostic user inputs and report an aggregate metric (Ouyang et al., 2022; Chung et al., 2022; Wang et al., 2023b; Dettmers et al., 2023). This approach has the potential to cover a wider range of use cases, however, it relies mostly on manual preference annotations from human labelers or larger LLMs which is costly and has no accepted protocol yet.

**D4. Knowledge of Data Origin.** So far, we discussed trust desiderata from the viewpoint of knowledge of functional capacity. Next to this, a model's behaviour is also largely influenced by its training data. Knowledge about data provenance helps us make informed decisions about whether a given LLM is a good match for the intended use case. Therefore, open access to data must be prioritized. In compartmentalized NLP, models are trained and evaluated on well-known, manually curated, task-specific datasets. Today's models are instead trained on task-heterogeneous corpora at web scale, typically of unknown provenance. For novel tasks, this means we do not know how well relevant facets (e.g., language, domain) are represented in the training data. For existing tasks, it is unclear if the model has seen test instances in their large training corpora (i.e., test data leakage; Piktus et al., 2023), blurring the lines between traditional

train-dev-test splits and overestimating the capabilities of LLMs. To compound matters further, models are not only trained on natural, but also on generated data, and unknown data provenance is also becoming an issue as annotators start to use LLMs (Veselovsky et al., 2023). LLMs trained on data generated by other LLMs can lead to a "curse of recursion" where (im-)probable events are over/underestimated (Shumailov et al., 2023).

## 3 What Can We Do to Gain Trust Now and in Future?

In a world where generative LLMs seemingly dominate every benchmark and are claimed to have reached human-level performance on many tasks,[3] we advocate that now is the time to treat trust as a first-class citizen and place it at the center of model development and evaluation. To operationalize the concept of trust, we denote with *knowledge facets* (henceforth, facets) all factors that improve our knowledge of functional capacity and knowledge of origin. Facets can be local (instance) or global (datasets, tasks). They refer to 1) descriptive knowledge such as meta-data or data/task provenance, and 2) inferred knowledge; for example which skills are exploited. We next propose concrete suggestions on how facets can help us gain trust in LLMs based on the desiderata in §2.

**Explain Skills Required versus Skills Employed.** It is instructive to think of prompt-based generative LLMs as instance-level problem solvers and, as such, we need to understand a-priori *the necessary skills for solving instances* (local facets) as well as knowing *what skills are actually employed during inference*. Most prior work aims to improve our understanding of tasks and the skills acquired to solve them by studying models trained specifically for each task, and can be broadly classified into: (i) linguistically motivated approaches and (ii) model-driven approaches (**D1**). Linguistic approaches formalize skills as cognitive abilities, which are studied, e.g., through probing tasks (Adi et al., 2017; Conneau et al., 2018; Amini and Ciaramita, 2023), checklists (Ribeiro et al., 2020) and linguistic profiling (Miaschi et al., 2020, 2021; Sarti et al., 2021). Model-driven approaches attribute regions in the model parameter space to skills (Ansell et al., 2022; Wang et al., 2022a; Ponti

---

[3]For example, GPT-4 reportedly passed the bar exam and placed top at GRE exams, see https://openai.com/research/gpt-4.

et al., 2023; Ilharco et al., 2023). The former can be seen as describing global facets (i.e., the overall functional capacity of black-box models), while the latter identifies local facets (i.e., skill regions in model parameters). To establish trust, we need to know what skills are required to solve instances, which is different from which skills are exercised by a model at inference time, as described next.

Besides knowlege about skills needed to solve a task, it is important to gain knowledge about what skills are actually being applied by an LLM. This is linked to explainability and transparency, corresponding to (i) understanding the knowledge[4] that goes into the inference process (**D1**), and (ii) the inference process itself in terms of applied skills (**D2**), e.g., examinations of LLMs' "thought processes". Regarding (i), existing work includes attributing training instances to model predictions (Pruthi et al., 2020; Weller et al., 2023) and explaining predictions through the lens of white-box models (Frosst and Hinton, 2017; Aytekin, 2022; Hedderich et al., 2022). They are, however, often grounded in downstream task data and thus do not provide insights connected to the knowledge memorized by LLMs during pre-training (*global facets*). Regarding (ii), existing approaches include guiding the generation process through intermediate steps (Wei et al., 2022c; Wang et al., 2023a; Li et al., 2023) and pausing the generation process to call external tools (Schick et al., 2023; Shen et al., 2023; Paranjape et al., 2023; Mialon et al., 2023). Their shortcoming is that they operate on the input level, and similarly do not capture cases where pre-existing, model-internal knowledge is applied. Furthermore, prior work has shown that LLMs follow the path of least resistance. That is, neural networks are prone to predict the right thing for the wrong reasons (McCoy et al., 2019; Schramowski et al., 2020), which can be caused by spurious correlations (Eisenstein, 2022).[5] On the path to gaining trust, we advocate for LLMs that are able to attribute their output to internal knowledge and the skills used to combine that knowledge. Alternatively, LLMs could be accompanied by white-box explanation models that (are at least a proxy) for explaining the inference process.

**Facilitate Representative and Comparable Qualitative Analysis.** Today, the standard target for

NLP papers proposing a new model is to beat previous models on a certain *quantitative* benchmark. We argue that if datasets and metrics are well-designed and well-grounded in skills/capabilities, they can be used as an indicator of progress.[6] On the other hand, findings from negative results might be obscured without *faceted quantitative analysis*: even when obtaining lower scores on a benchmark, sub-parts of an NLP problem may be better solved compared to the baseline, but go unnoticed (**D3**). We therefore cannot trust reported SOTA results as long as the facets that explain how well sub-problems are solved remain hidden. Complementary to holistic quantitative explanations, as proposed by HELM (Liang et al., 2022), we call for a holistic qualitative evaluation where benchmarks come with *standardized qualitative evaluation protocols*, which facilitates comparable qualitative meta-analysis. This proposal is inspired by the manually-curated GLUE diagnostics annotations (Wang et al., 2018), which describe examples by their linguistic phenomena.[7] Recycling existing tasks and augmenting them with diagnostic samples to study LLMs provides a very actionable direction for applying existing compartmentalization in a more targeted trustworthy way. Diagnostics samples should ideally represent the full spectrum of cognitive abilities required to solve a task. Designing these samples is however a complex task. We hypothesize that the set of required skills varies between tasks and should ideally be curated by expert annotators.

**Be Explicit about Data Provenance.** In ML, it is considered good practice to use stratified data splits to avoid overestimation of performance on dev/test splits based on contamination. Traditionally, this stratification was done based on, e.g., source, time, author, language (cross-lingual), or domain (cross-domain). Recent advances have hinted at LLMs' ability to solve new tasks, and even to obtain new, i.e., emergent abilities (Wei et al., 2022b). These are in fact similar cross-$\mathcal{X}$ settings, where $\mathcal{X}$ is no longer a property at the level of dataset sampling, but of the broader task setup. We call for always employing a cross-$\mathcal{X}$ setup (**D4**); whether it is based on data sampling, tasks, or capabilities— urging practitioners to make this choice explicit. Transparency about data provenance and test data

---

[4]Including acquired knowledge such as common sense and world knowledge (Li et al., 2022b; De Bruyn et al., 2022).

[5]"The sentiment of a movie should be invariant to the identity of the actors in the movie" (Eisenstein, 2022)

---

[6]Note that baseline comparisons can still be obscured by unfair comparisons (Ruffinelli et al., 2020).

[7]https://gluebenchmark.com/diagnostics/

leakage improve our trust in reported results. In practice, these data provenance facets are also valuable for identifying inferred knowledge such as estimated dataset/instance difficulty (Swayamdipta et al., 2020; Rodriguez et al., 2021; Ethayarajh et al., 2022), especially when used in conjunction with the aforementioned diagnostic facets.

Data provenance is also important when drawing conclusions from benchmark results (**D3**). Tedeschi et al. (2023) question the notion of superhuman performance and claims of tasks being solved (i.e., overclaiming model capabilities), and criticize how benchmark comparisons "do not incentivize a deeper understanding of the systems' performance". The authors discuss how external factors can cause variation in human-level performance (incl. annotation quality) and lead to unfair comparisons. Similarly, underclaiming LLMs' capabilities also obfuscates our knowledge of their functional capacity (Bowman, 2022). Additionally, in a recent study domain experts find the accuracy of LLMs to be mixed (Peskoff and Stewart, 2023). It is therefore important to be explicit about the limitations of benchmarks (Raji et al., 2021) and faithful in communicating model capabilities. At the same time, it is an ongoing discussion whether reviewers should require (i.e, disincentivize the absence of) closed-source baseline models such as ChatGPT and GPT-4, which do not meet our trust desiderata (Rogers et al., 2023). Closed-source models that sit behind APIs typically evolve over time and have unknown data provenance, thus lacking both knowledge of origin (**D4**), and the consistency of its functional capacity. Consequently, they make *untrustworthy baselines* and should not be used as an isolated measure of progress.

## 4   Trustworthiness and User Trust

So far we have discussed different avenues for improving our knowledge about LLM's functional capacity and origin, paving the way for establishing trustworthiness. From a user perspective it is essential to not only understand knowledge facets but also how they empirically impact *user trust* in a collaborative environment. This is especially important in high-risk scenarios such as in the medical and legal domain. One could argue, if LLMs such as ChatGPT are already widely adopted, do we already trust LLMs (too much)? To better understand user trust we need interdisciplinary research and user experience studies on human-AI collaboration.

Specifically, we need to know what users do with the model output across multiple interactions (e.g., verify, fact check, revise, accept). For example, González et al. (2021) investigate the connection between explanations (**D2**) and user trust in the context of question answering systems. In their study users are presented with explanations in different modalities and either accept (trust) or reject (don't trust) candidate answers. Similarly, Smith-Renner et al. (2020) discuss how generated explanations can promote over-reliance or undermine user trust. A closely related question is how the faithfulness of explanations affect user trust (Atanasova et al., 2023; Chiesurin et al., 2023). For a comprehensive overview on user trust we refer to the recent survey by Bach et al. (2022).

While such controlled studies using human feedback are cost and time intensive, the minimum viable alternative for establishing trust may simply be the publication of a model's input-output history. In contrast to standalone metrics and cherry-picked qualitative examples, access to prior predictions enables post-hoc *knowledge of model behaviour* (**D2**), even without direct access to the model. This democratizes the ability to verify functional capacity and helps end users seeking to understand how well a model works for their task.

In summary, evaluating user trust is an integral part of trustworthiness and goes hand in hand with careful qualitative analyses and faceted quantitative evaluation. Towards this goal, we believe LLM development needs to be more human-centric.

## 5   Conclusions

In this position paper, we emphasize that the democratization of LLMs calls for the need to rethink tasks and model evaluation, placing trustworthiness at its center. We adopt a working definition of trustworthiness and establish desiderata required to improve our knowledge of LLMs (§2), followed by suggestions on how trust can be gained by outlining directions guided by what we call *knowledge facets* (§3). Finally, we draw a connection between trustworthiness as knowledge facets and user trust as means to evaluate their impact on human-AI collaboration (§4).

## Limitations

To limit the scope of this work, we did not discuss the topics of social and demographic biases (Gira et al., 2022), discrimination of minority groups

(Lauscher et al., 2022) and hate speech as factors influencing our trust in LLMs. Within our proposed desiderata, this facet would fall under 'Knowledge of Data Origin' (§2), in terms of understanding where model-internal knowledge and the associated biases originate from (**D4**).

Our proposed multi-faceted evaluation protocols rely strongly on human input—either via qualitative judgements and/or linguistically annotated diagnostic benchmarks (§3). We acknowledge that such analyses require more time and resources compared to evaluation using contemporary, automatic metrics, and may slow down the overall research cycle. While we believe that slower, yet more deliberate analyses are almost exclusively beneficial to establishing trust, our minimum effort alternative of publishing all model predictions can also be used to build user trust (§4). This simple step closely mirrors the scientific method, where hypotheses must be falsifiable by anyone (Popper, 1934). Identifying even a single incorrect prediction for a similar task in a model's prediction history, can already tell us plenty about the model's trustworthiness.

## Acknowledgements

We thank the anonymous reviewers for their insightful comments. This research is supported by the Independent Research Fund Denmark (DFF) Sapere Aude grant 9063-00077B and ERC Consolidator Grant DIALECT 101043235.

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
