# OpenReview forum: "Establishing Trustworthiness: Rethinking Tasks and Model Evaluation"
_EMNLP/2023/Conference — EMNLP 2023 Main_

### Official Review · Reviewer_8KkL · 2023-08-02

**Soundness:** 4

**Excitement:**

4: Strong: This paper deepens the understanding of some phenomenon or lowers the barriers to an existing research direction.

**Missing References:**

In section 3, it feels like the recent work on chain-of-thought and improving the logical consistency of CoT is relevant, for example:
- Jung, Jaehun, et al. "Maieutic Prompting: Logically Consistent Reasoning with Recursive Explanations." Proceedings of the 2022 Conference on Empirical Methods in Natural Language Processing. 2022.

**Paper Topic And Main Contributions:**

This position paper outlines current issues of applying standard NLP task and evaluation paradigms to the new generation of large LMs (GPT-4 etc.). Using David Hays' definition of trustworthiness, they outline 4 desiderata for trustworthy models: knowledge of inputs, knowledge of behavior, knowledge of evaluation protocols, and knowledge of data origins. Based on these, authors discuss some avenues for improving trustworthiness of NLP systems.

**Questions For The Authors:**

At the risk of being somewhat philosophical, but my understanding is that the paper rests on the premise that NLP systems should be explainable, but the word "trustworthy NLP systems" doesn't feel like the right goal, given that lay humans trust AI a lot already (too much?), despite having no idea how it works. What are the authors' thoughts on that distinction?

**Reasons To Accept:**

- This paper presents a solid outline of how the shift to instruction-tuned NLP systems requires new evaluation paradigms.
- The use of Hays' definition and the four desiderate described make sense to me.
- The listed directions and recommendations make sense to me.

**Reasons To Reject:**

- The main reason to reject is the following: while I agree and appreciate most of the arguments made in the paper and future directions towards more trustworthy NLP, the paper's impact and usefulness for the community could be improved by making the recommendations more concrete and fleshed out.
  - For example, on L277, this feels like an ambitious and complex thing to ensure that LLMs be able to attribute their outputs to internal knowledge... How do authors believe this could be done? Or at least, what are some general directions towards that?
  - Another example, on L306, authors suggest the use of diagnostic samples to study LLMs behavior, which I love; but how should NLP practitioners actually craft these? Is there a framework for crafting these? What is some prior work that has done something similar, that practitioners could rely upon?
- A second concern is the lack of discussion around incentive structures of current landscape of NLP research, including industry profit and risks of harms. Specifically, there are several reasons why the current big players of LLMs have directly acknowledged that transparency is antithetical to their profit incentives (e.g., the OpenAI GPT-4 report describes the training data details as OpenAI's own "intellectual property," as a reason for not sharing those details). Additionally, some of these desiderata have their own potential risks (for example, releasing an entire training dataset for an LLM could include releasing copyrighted materials, or issues of data sovereignty where data is in languages that should be stewarded only by folks who speak those languages). I wish authors could at least acknowledge these incentives and potentially even mention ways to change these incentives.

**Edit: post-rebuttal update:** Changed my scores from 3/3 to 4/4

**Reproducibility:**

N/A: Doesn't apply, since the paper does not include empirical results.

**Reviewer Confidence:**

4: Quite sure. I tried to check the important points carefully. It's unlikely, though conceivable, that I missed something that should affect my ratings.

---

> ### Author Rebuttal · Authors · 2023-08-28
>
> Thank you very much for taking the time to write such a thorough review and for sharing your thoughts. We interpret this as a positive sign given that one of our main goals of this position paper is to stimulate discussions in the research community. Please find below our replies to your specific concerns and questions, we will use the extra space in the final version of the paper to incorporate your valuable feedback.
>
> Regarding your questions on L277, we acknowledge attributing model output to internal knowledge and skills used is a challenging goal, and only one step towards a multi-faceted evaluation, yet an important one. We believe attribution requires a deeper understanding of model behavior, including (but not limited to) localizing skills and knowledge in the model’s parameter space. Prior work towards this goal has been done in the context of localizing internal factual knowledge [1-4] and localizing task-specific parameter regions (L235-237). Extending this to other types of knowledge (e.g., common sense knowledge, social biases) presents opportunities for future work, as well as opportunities to uncover what can and what cannot be done with attribution methods, therefore we deem it an important facet. Furthermore, viewing tasks as mixtures of skills allows future work in another general direction, namely extending methods for task representations towards the skill and instance-level. Thank you for your comment, we will make sure to be more specific on the general directions.
>
> Regarding your question on L306, thank you for your positive feedback and very good question. Generally, diagnostics samples should ideally represent the full spectrum of required cognitive abilities required to solve a task. This is however a complex task. We hypothesize that the set of required skills varies between tasks. Representative test examples and probes should be manually curated by expert annotators (L353-356) as opposed to cherry-picked test examples (L365), but the crafting remains an open problem. Here, we will add Raji et al. [7] to point out the limitations of benchmarking to measure general language understanding capabilities. As there exists no framework, we call for more work in this direction. Recent work on collecting explanations on ambiguous and uncertain cases (e.g. [6]) can be taken as an actionable starting point.
>
> Regarding incentive structures: Thank you for raising this important point. We agree that the manuscript benefits from explicitly touching upon current incentive structures. We will add a discussion (e.g. expand D.4 with the current lack of incentives on data transparency, and extend the discussion in L356 with examples like the transparency issue). We believe that adding this helps the discussion on how incentives structures can be changed and we thank the reviewer for this constructive feedback. For instance, we believe that rethinking tasks and model evaluation with trustworthiness in mind can be a valuable conceptual framework to revisit incentive structures. For example, an ongoing discussion is whether reviewers should require (i.e, disincentivize the absence of) closed-source baseline models such as ChatGPT and GPT-4, which don’t meet our trust desiderata [5]. In addition, there is a lot to gain from rethinking benchmarks, slowing down research (L356-362) and we call for incentivizing more elaborate evaluation protocols and systematic qualitative evaluation (L299-303).
>
> Regarding the Question: Thank you for this insightful question! We would like to clarify that explainability alone is not sufficient, our perspective on trust and our trust desiderata go beyond current explainability methods and beyond the users' perspective. For example, we argue that we cannot trust reported results in the absence of data provenance information (L290-297, L326-327). Overall, we believe however that it is important to  make the crucial distinction between the widespread adoption of LLMs and the user’s trust in those models. Whether end users trust (too much?) LLMs, and whether and how more trustworthy systems impact user trust, is an intriguing question. To get a better understanding of this phenomenon we need interdisciplinary collaboration and user experience studies (human-AI collaboration). Specifically, to answer the question about user trust we would need to know what users do with the model output across multiple interactions (longitudinal, e.g., verify, fact check, revise, accept) and do, e.g., user interviews. An interesting research question given our desiderata could be: How does user trust compare to systems that implement our desiderata vs. systems that do not (control group). We will include this discussion on the distinction between trustworthiness and user trust in the final version.
>
> Thank you for the suggested paper reference, we agree that CoT literature is indeed relevant to our work (L262-265). We will add it to the paper, together with the references discussed here.
>
> [1] “Locating and Editing Factual Associations in GPT” Meng et al. (NeurIPS’22)\
> [2] “Mass-Editing Memory in a Transformer” Meng et al. (ICLR’23)\
> [3] “Editing Factual Knowledge in Language Models” De Cao et al. (EMNLP’22)\
> [4] “When Not to Trust Language Models: Investigating Effectiveness of Parametric and Non-Parametric Memories” Mallen et al. (ACL’23)\
> [5] “Closed AI Models Make Bad Baselines” Anna Rogers (Blog Post, 2023)\
> [6] "Understanding and Predicting Human Label Variation in Natural Language Inference through Explanation" Jian & de Marneffe (2023, arXiv)\
> [7] "AI and the Everything in the Whole Wide World Benchmark" Raji et al., (NeurIPS 2021)

---

### Official Review · Reviewer_kEVc · 2023-08-04

**Soundness:** 3

**Excitement:**

3: Ambivalent: It has merits (e.g., it reports state-of-the-art results, the idea is nice), but there are key weaknesses (e.g., it describes incremental work), and it can significantly benefit from another round of revision. However, I won't object to accepting it if my co-reviewers champion it.

**Paper Topic And Main Contributions:**

This paper emphasizes that the democratization of LLMs calls for the need to rethink tasks and model evaluation, with a focus on placing trustworthiness at the center of these considerations.

**Reasons To Accept:**

--This paper synthesizes contemporary work distributed throughout different subfields of NLP and ML into a conceptual framework for trust.

--This paper provides a different and valuable perspective for analyzing LLMs.


**Reasons To Reject:**

--The authors point out that recent LLMs break down the compartmentalization (dashed lines) in Figure 1, impacting all stages of the cycle. Can a corresponding Figure be provided to offer readers a more visually informative comparison between the two?

--In Section 2, each trust desideratum should be accompanied by its corresponding explanation, making it easier for readers to understand. For example, "Knowledge about Model Input refers to..."

--In Section 3, the conclusions drawn from the analysis of existing work, which provide insightful implications for future research, should be presented more explicitly and clearly.


**Reproducibility:**

2: Would be hard pressed to reproduce the results. The contribution depends on data that are simply not available outside the author's institution or consortium; not enough details are provided.

**Reviewer Confidence:**

2: Willing to defend my evaluation, but it is fairly likely that I missed some details, didn't understand some central points, or can't be sure about the novelty of the work.

---

> ### Author Rebuttal · Authors · 2023-08-28
>
> Regarding Weakness 1: Thank you for this suggestion. In Figure 1 we show the traditional NLP compartments with clearly defined evaluation protocols, task definitions and datasets, indicated with dashed lines, effectively describing the “before” state. In the current state of NLP those boxes tend to disappear as datasets are combined and tasks and evaluation protocols are conflated. We will update the diagram to better highlight the paradigm shift from NLP compartments towards general purpose, task-agnostic approaches (L008-L012).
>
> Regarding Weakness 2: Given the extra space in the final version, we will improve the overall clarity of each desiderata. In particular, in D1 we refer to knowledge about “ingredients” that goes into the model’s decision making (L104-107, Footnote 2), this includes unwanted ingredients such as social biases and propaganda (L344-348). In D2 we refer to knowledge about the skills that are used to process those ingredients into model outputs, e.g., by being able to interpret decision boundaries (L130), this includes unwanted skills learned from spurious correlations (L275). D3 and D4 refer to our knowledge that help us gauge the true capability of models.
>
> Regarding Weakness 3: Thank you for the feedback. We will improve the overall clarity of our paper and also highlight the implications on future research more explicitly by including a discussion on incentive structures and concrete examples of possible research directions. Please refer to our reply to 8KkL’s question and our response regarding suggestions on attributing model output to internal knowledge, building diagnostics datasets and conducting user trust studies.

---

### Official Review · Reviewer_H6uV · 2023-08-05

**Soundness:** 4

**Excitement:**

4: Strong: This paper deepens the understanding of some phenomenon or lowers the barriers to an existing research direction.

**Paper Topic And Main Contributions:**

This paper argues that with the rise of large language models (LLMs), it is becoming increasingly difficult to establish trust in their capabilities since the traditional NLP handle compartmentalized tasks and evaluations but LLMs are applied extremely broadly. It proposes to place trust at the center of model development and evaluation, and rethink task formalization and evaluation around the concept of "knowledge facets" - factors that improve understanding of a model's origins and functional capacities. This paper outlines desiderata for trustworthy LLMs, including knowledge of model inputs, behavior, evaluation protocols, and data origin. It also proposes some potential solutions, including to explain required versus employed skills, facilitate qualitative analysis, and be explicit about data provenance.

**Questions For The Authors:**

Question A. In the "Explain Skills Required versus Skills Employed" part of Section 3, I understand that the first paragraph is about "Skills required" and the second paragraph is about "Skills Employed". Why do you discuss the approachs attributing regions in the model parameters as identifing "Skills Required" rather than "Skill Employed"? I thought they are finding which neurons/skills are activated/employed during task solving.


**Reasons To Accept:**

(1) This paper performs excellently as a position paper. The introduction's discourse on why the emergence of Large Language Models (LLMs) has made trustworthy NLP more important is appropriate and convincing. The analysis of the desiderata is also well executed.

(2) Not only does this article provide new insights and proposals, but it also comprehensively surveys and organizes existing work within a limited space. I believe that by following the framework provided by this paper to read the cited works, readers will gain a deeper understanding of the research landscape.

**Reasons To Reject:**

(1) The part of proposed actions (Section 3) is not clear and specific enough to me. It would be better to use more space and add some examples or illustrations for them. It will also be helpful to explicitlt split the parts about background and related work between the parts of proposed actions. Also, the proposal in the Limitations section is interesting and I think it should be placed into the content. This position paper might be better to be presented in long paper.

**Reproducibility:**

N/A: Doesn't apply, since the paper does not include empirical results.

**Reviewer Confidence:**

4: Quite sure. I tried to check the important points carefully. It's unlikely, though conceivable, that I missed something that should affect my ratings.

**Typos Grammar Style And Presentation Improvements:**

(1) Line 306: duplicate "linguistic".
(2) Line 340: missing preposition between "knowledge" and "LLMs".
(3) Some references are duplicate, such as Wei et al., 2022a and 2022b, and Touvron et al., 2023a and 2023b.

---

> ### Author Rebuttal · Authors · 2023-08-28
>
> Thank you for your concrete suggestions, and the detailed review of our work. We are highly encouraged by your positive feedback regarding the execution of our desiderata analysis.
>
> Regarding Weakness 1: We fully agree that the best use of additional space would be to be more concrete on explaining desiderata, please also refer to our response to Reviewer 8KkL. One of our primary goals is to propose trust desiderata as a thought framework for stimulating and guiding discussions in the community. To this end, we hope to inspire possible research directions for future uptake. We also agree that parts of the Limitations would fit also in the main content. Given the extra space in the final version we will revisit the paper structure.
>
> Regarding Question A: This indeed appears to be an error in the writeup, we will revise the paper in the final version. Thank you for highlighting this. We also appreciate the concrete suggestions for presentation improvement.

---

### Meta-Review · Area_Chair_CCLV · 2023-09-19

**Recommendation:** 4

**Metareview:**

This position paper investigated an important question, i.e. how should we evaluate LLMs. Using David Hays' definition of trustworthiness, the authors outline 4 desiderata for trustworthy models. The authors discuss some avenues for improving the trustworthiness of NLP systems. It is basically an excellent position paper and well motivated by the introduction section. The analysis of the desiderata is also well executed. It also comprehensively surveys and organizes existing work within a limited space.

---

### Decision · Program_Chairs · 2023-10-07

**Decision:**

Accept-Main

**Comment:**

This position paper investigated an important question, i.e. how should we evaluate LLMs. Using David Hays' definition of trustworthiness, the authors outline 4 desiderata for trustworthy models. The authors discuss some avenues for improving the trustworthiness of NLP systems. It is basically an excellent position paper and well motivated by the introduction section. The analysis of the desiderata is also well executed. It also comprehensively surveys and organizes existing work within a limited space.